

# *Sida chlorotic leaf virus*: a new recombinant begomovirus found in non-cultivated plants and *Cucumis sativus* L

Daniel Alejandro García-Rodríguez[1,*], Brenda Lizet Partida-Palacios[1,*], Carlos Fernando Regla-Márquez[2], Sara Centeno-Leija[3], Hugo Serrano-Posada[3], Bernardo Bañuelos-Hernández[4] and Yair Cárdenas-Conejo[3]

[1] Universidad de Colima, Laboratorio de Biología Sintética, Estructural y Molecular, Colima, México
[2] Centro Nacional de Referencia Fitosanitaria, Departamento de Control Biológico, Colima, México
[3] Universidad de Colima, Consejo Nacional de Ciencia y Tecnología-Laboratorio de Biología Sintética, Estructural y Molecular, Colima, México
[4] Escuela de Veterinaria, Universidad de La Salle, Bajío, León, Guanajuato, México
[*] These authors contributed equally to this work.

Corresponding author
Yair Cárdenas-Conejo,
ycardenas11@ucol.mx

## ABSTRACT

**Background**. Begomoviruses are circular single-stranded DNA plant viruses that cause economic losses worldwide. Weeds have been pointed out as reservoirs for many begomoviruses species, especially from members of the *Sida* and *Malvastrum* genera. These weeds have the ability to host multiple begomoviruses species simultaneously, which can lead to the emergence of new viral species that can spread to commercial crops. Additionally, begomoviruses have a natural tendency to recombine, resulting in the emergence of new variants and species.

**Methods**. To explore the begomoviruses biodiversity in weeds from genera *Sida* and *Malvastrum* in Colima, México, we collected symptomatic plants from these genera throughout the state. To identify BGVs infecting weeds, we performed circular DNA genomics (circomics) using the Illumina platform. Contig annotation was conducted with the BLASTn tool using the GenBank nucleotide "nr" database. We corroborated by PCR the presence of begomoviruses in weeds samples and isolated and sequenced the complete genome of a probable new species of begomovirus using the Sanger method. The demarcation process for new species determination followed the International Committee on Taxonomy of Viruses criteria. Phylogenetic and recombination analyses were implemented to infer the evolutionary relationship of the new virus.

**Results**. We identified a new begomovirus species from sida and malvastrum plants that has the ability to infect *Cucumis sativus* L. According to our findings, the novel species *Sida chlorotic leaf virus* is the result of a recombination event between one member of the group known as the Squash leaf curl virus (SLCV) clade and another from the Abutilon mosaic virus (AbMV) clade. Additionally, we isolated three previously identified begomoviruses species, two of which infected commercial crops: okra (*Okra yellow mosaic Mexico virus*) and cucumber (*Cucumber chlorotic leaf virus*).

**Conclusion**. These findings support the idea that weeds act as begomovirus reservoirs and play essential roles in begomovirus biodiversity. Therefore, controlling their populations near commercial crops must be considered in order to avoid the harmful effects of these phytopathogens and thus increase agricultural efficiency, ensuring food and nutritional security.

# INTRODUCTION

The *Geminiviridae* family encompasses several phytopathogen viruses globally relevant due to the wide range of hosts and vectors in tropical and subtropical regions (*Rojas et al., 2005a*). These viruses possess an icosahedral capsid and small single-stranded circular DNA (2.5–5.2 kb) that is replicated through the rolling circle replication (RCR) mechanism (*Jeske, 2009*). So far, nearly 520 species have been described with the potential to infect both dicotyledonous and monocotyledonous plants. The *Geminiviridae* family is divided into 14 genera (*Becurtovirus, Begomovirus, Capulavirus, Citlodavirus, Curtovirus, Eragovirus, Grablovirus, Maldovirus, Mastrevirus, Mulcrilevirus, Opunvirus, Topilevirus, Topocuvirus*, and *Turncurtovirus)* based on their unique vectors, genomic organization, and hosts (*Fiallo-Olivé et al., 2021*).

The genus *Begomovirus* is the most diverse group of geminiviruses, with 445 species reported and their potential to cause significant damage to crops of commercial interest globally (*Fiallo-Olivé et al., 2021*) by hindering the quality and quantity of the affected crops (*Bhattacharjee & Hallan, 2022*). There have been cases where these viruses have caused low yields and complete losses in the crops they infect (*Vu, Roy Choudhury & Mukherjee, 2013*). Some remarkable examples of this have occurred around the world: the African cassava mosaic virus is responsible for losses ranging from 20 to 95% in Africa. In Brazil, the species *Bean golden mosaic virus* was responsible for the most devastating event, for that country, generating losses ranging up to 75% in the affected crops (*Fauqet & Fargette, 1990*; *Vu, Roy Choudhury & Mukherjee, 2013*). Begomoviruses (BGVs) mainly affect tropical and subtropical regions worldwide, and are spread by the whitefly *Bemisia tabaci* (*Hemiptera; Aleyrodidae*) from plant to plant. Another remarkable feature is that the *Begomovirus* genus is comprised of species with one genomic component (monopartite BGVs) or two genomic components (bipartite BGVs) (*Zerbini et al., 2017*). Except for a 200-base long intergenic region (IR) known as the common region (CR), the genomic components A (DNA-A) and B (DNA-B) do not retain pairwise identity. This segment of the genomes holds the viral replication origin (Ori) formed by iterons and a secondary structure where an ultra-conserved nicking site (5′-TATAATATT/AC-3′) in BGVs is maintained (*Argüello-Astorga et al., 1994*; *Hanley-Bowdoin et al., 2000*). The genome organization of monopartite BGVs and component A of bipartite BGVs are similar. In the virion sense are found the genes coding for the capsid protein (CP) and the movement protein (MP), the last one is characteristic of old-world BGVs. The open reading frames of the replication-associated protein (Rep), replication enhancer protein (REn), transcription activator protein (TrAP), and the C4/AC4 protein are held in the complementary sense and have been linked to gene silencing regulation events within the virus-host (*Lazarowitz, 1992*; *Jeske, 2009*). Besides, component B is focused on the BGVs movement, and here we can find genes for the nuclear shuttle protein (NSP) and movement protein (MP) that favors

the intracell and extra cell transport, respectively (*Lazarowitz, 1992*; *Pascal et al., 1994*). Diverse phylogeny analyses subdivide the BGVs into four groups: New World (NW), Old World (OW), legumoviruses, and sweepoviruses (*Ilyas et al., 2009*; *Briddon et al., 2010*). Most OW BGVs (native to Africa, the Indian subcontinent and Asia) are monopartite and commonly associated with other extrachromosomal DNAs known as satellites. NW BGVs are mostly bipartite (native to the Americas), and satellites are rarely associated with them. The NW BGVs have been grouped into multiple clades where the Squash leaf curl virus clade (SLCV) clade, Abutilon mosaic virus (AbMV) clade, and Brazilian clades outstand (*Argüello-Astorga et al., 1994*; *Rojas et al., 2005b*).

In agriculture, the adverse effects of weeds are well-known. They can directly produce adverse outcomes such as reduced yield due to nutrient competition or parasitism with the desired crop (*Scavo & Mauromicale, 2020*). Furthermore, weeds have a remarkable potential to act as reservoirs or alternative hosts for multiple BGVs species, facilitating the dispersion of BGVs on crops of commercial interest through vector-based transmission (*Jovel et al., 2004*; *McLaughlin et al., 2008*). Also, multiple infections of weeds by begomoviruses can lead to recombination events that could originate new BGVs species, as this is a common mechanism used by these viruses (*Padidam, Sawyer & Fauquet, 1999*; *Jovel et al., 2004*; *Rojas et al., 2005b*; *Graham, Martin & Roye, 2009*; *Tavares et al., 2012*; *Ferro et al., 2017*). Weeds from the *Sida* and *Malvastrum* genera are particularly relevant hosts due to their function as reservoirs and are natural hosts for 37 and 11 species, respectively, making them the group of weeds that harbor the most begomoviral species (*Stewart et al., 2014*; *Alabi et al., 2016*; *Fiallo-Olivé et al., 2021*; *Chahwala et al., 2021*). Therefore, plants from the genera *Sida* and *Malvastrum* pose a risk to agriculture as they can act as incubators for new BGVs species due to the frequent coexistence of diverse viruses within the same host (*Jovel et al., 2004*). Additionally, some BGVs species identified from sida and malvastrum weeds can migrate to commercial crops such as soybean (*Fernandes et al., 2009*), bean (*Durham et al., 2010*), tomato (*Rocha et al., 2013*), cucumber (*Sanchez-Chavez et al., 2020*).

México possesses an extraordinary natural wealth that positions it among the 17 megadiverse countries; 10–12% of all global biodiversity can be found in this country (*Sarukhán et al., 2009*; *Cruz-Angón & Perdomo-Velázquez, 2016*). Additionally, México has high weather variability due to its orography and geographical localization in the transition zone between the tropical and temperate regions of the planet (*Vidal-Zepeda, 2005*). These features are favorable for the proliferation of *B. tabaci* (*Zerbini et al., 2017*) and multiple BGVs species that affect the production of crops such as tomato, cucumber, bean and soybean. Currently, 35 species have been reported in México (*Mauricio-Castillo et al., 2007*; *Gregorio-Jorge et al., 2010*; *Brown et al., 2015*). In the past years, Mexican agriculture has suffered devastating effects from BGVs; for example, the tomato yellow leaf curl virus caused crop losses valued at 300 million dollars in 2006 (*Brown & Idris, 2006*).

Colima is one of the 31 states of México, located on the west coast (19°05′48″N 103°57′39″O) in the tropical area of the country. Despite its small size (only 0.3% or 5,627 km$^2$ of México), it has vast biodiversity with varied physiography and climatology that subserves the existence of multiple ecosystems harboring a high number of living beings (*Cruz-Angón & Perdomo-Velázquez, 2016*). In this state, 22% of the begomoviral species

identified in México have been isolated (*Morales et al., 2005*; *Rodríguez-Negrete et al., 2019*; *Sanchez-Chavez et al., 2020*).

The primary objective of this study was to explore the biodiversity of begomoviruses within the genera of weeds that host the highest number of begomoviral species (*Malvastrum* and *Sida*). As a result, we discovered a new BGV species in both sida and malvastrum weeds that exhibited the ability to infect cucumber plants. Additionally, our research led us to conclude that this virus is a result of a recombination event between one member of the AbMV clade and the species *Cucumber chlorotic leaf virus* (CuChLV), a BGV recently isolated in a cucumber field in Colima, México (*Sanchez-Chavez et al., 2020*). Our findings reinforce the idea that weeds play an important agricultural and evolutive role because they serve as reservoirs and incubators for new varieties and species of BGVs. Therefore, their control is essential to prevent negative impacts on commercial crops caused by BGVs infections.

## MATERIALS & METHODS

### Sample collection and DNA extraction

Leaves with typical geminivirus disease symptoms from the *Sida* and *Malvastrum* genera (Fig. 1) were collected from Colima, México (19.2209, −103.8000; 19.2049, −103.7557; 19.3086, −103.5553; 19.3335, −103.7373; 19.3618, −103.7945) during August 2018. The DNA extraction for the 30 collected samples was carried out with the Dellaporta DNA extraction method (*Dellaporta, Wood & Hicks, 1983*) with slight modifications that include one step to denature and separate the proteins from the DNA using a phenol-chloroform isoamyl alcohol mixture. To enrich the viral genomes, total DNA extraction from each plant sample was used as a template for rolling circle amplification (RCA) using phi29 DNA polymerase (Thermo Fisher Scientific, Waltham, MA USA). The protocol was performed according to *Bhat & Rao (2020)*. RCA/restriction fragment length polymorphism (RFLP) was performed to select geminivirus-positive samples for sequencing. Positive samples are considered to be those that showed patterns in which band sizes added up to multiples of 2,600 nt. For the RCA/RFPL protocol, *Nde* I, *Xba* I and *Xho* I restriction enzymes were used. 3 μg of RCA products from each plant were pooled, then 1 μg was used as a template for NGS.

### Circular DNA genomics (circomics) by Illumina

The pooled samples were sent to the Laboratory of Genomic Services (LABSERGEN, Irapuato, Guanajuato, México) to be sequenced. A library for Illumina sequencing was prepared with the TrueSeq DNA Nano kit (Illumina Inc., San Diego, CA, USA). The library was paired-end (PE) sequenced with 150 cycles in one lane of the Illumina NextSeq 500 platform (∼1 million reads), using fragments of 480 bp in length. Bcl2fastq2 Conversion software v2.19.1 (Illumina Inc., San Diego, CA, USA) was used to convert raw data to fastq files and remove adapters. Filtering of low quality reads was performed with Trimmomatic v0.38 (*Bolger, Lohse & Usadel, 2014*). Reads were retained if the read length was ≥50 bp and the average Phred33 sequence quality (over a 5 bp window) was above 20. Quality control was carried out with FastQC v0.11.8-0 and MultiQC v1.6, and the reads were assembled
(a)  (b)

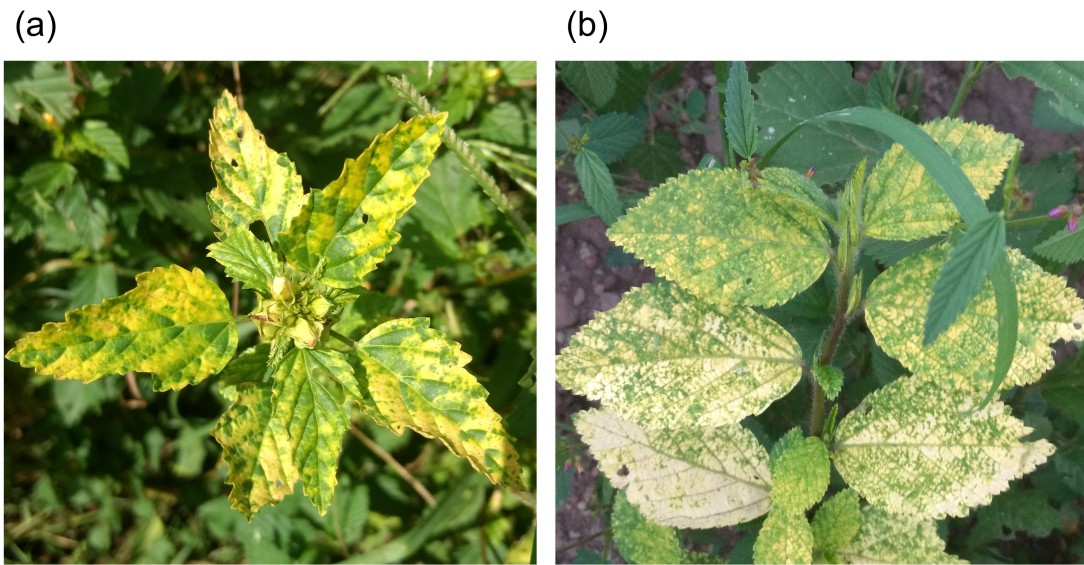

**Figure 1 Representative symptoms of collected weeds from Colima State México.** Weeds from *Malvastrum genus* displayed yellow mosaic (A), and most weeds from *Sida* genus showed yellow sprinkles and chlorosis (B).

with SPAdes v3.12.0 using default settings (*Bankevich et al., 2012*). Finally, CAP3 was used as a super assembler (*Huang & Madan, 1999*) to extend SPAdes contigs. Contigs smaller than 250 bp were filtered out. Assemblathon v2 (*Bradnam et al., 2013*) was used to calculate the mean contig size and N50.

## Geminivirus identification

To identify geminiviruses, a search was performed against the nucleic acid sequence database (GenBank: *nt* database; updated on March 2021), using the BLASTn algorithm included in the bioinformatics package BLAST+ v2.11.0 (*Zhang et al., 2000*). Hits with an e-value greater than $1e^{-20}$ were discarded. Contigs with positive hits to geminivirus were selected and manually arranged to get sequences in a virion sense, and the nicking site was established as the nucleotide number one. Finally, curated contigs were again compared with the RefSeq viral database (April 2021) to confirm the identification of geminiviruses.

## Sanger sequencing to corroborate the genome sequences of the new virus

PCR analysis was carried out using primers designed for each virus (Table S1) to confirm the presence of BGVs identified by the metagenome. PCR reaction was performed in C1000 Thermal cycler (BioRad, Hercules, CA, USA) as previously described in *Sanchez-Chavez et al. (2020)*. For the new virus, amplicons were inserted in pGEM T-easy vector (PROMEGA, Madison, WI, USA) and sequenced at LANBAMA-IPICYT (San Luis Potosí, México) using a 3130 Genetic Analyzer (Applied Biosystems, Waltham, MA, USA) for Sanger sequencing.

## Species demarcation

The demarcation analysis proposed by *Brown et al. (2015)* was carried out to determine if a new species of begomovirus was identified. First, using the BLASTn tool, we searched for related BGVs, using the possible new virus as a query, in the *nt* database with the search term "txid10814". Second, the first 250 hits were selected, and the complete DNA-A sequences were downloaded and curated to establish the nicking site as the nucleotide number one. Third, the DNA data set was aligned employing the MUSCLE algorithm contained within the package SDT v1.2 (freely available at http://www.cbio.uct.ac.za/SDT) to calculate identities between every pair of sequences.

## Infectivity test of sida chlorotic leaf virus in *Nicotiana benthamiana*

For the plant infectivity test, *N. benthamiana* plants were grown in the climatic chamber CLIMACELL 707 (MMM Group, Munich, Germany) at 28 °C with a photoperiod of 16 h of light/8 h dark and approximately 65% humidity. Infectious clones of DNA-A and DNA-B were agroinoculated to four week old *N. benthamiana* plants following the Clemente protocol (*Clemente, 2006*). To construct the infective clone of DNA-A, the origin of replication was cloned in the pBlueScript II KS+ vector (Addgene, Watertown, MA, USA) at *Sma* I and *Pst* I sites. This construction was linearized with *Pst* I and the complete component A inserted, generating 1.5 copies of DNA-A. Finally, the 1.5 mer of DNA-A was extracted from the pBlueScript II KS+ vector using the *Pvu* II enzyme and then subcloned into the pBI121 vector at the *Sma* I restriction site. On the other hand, the 1.5 mer of DNA-B was generated similarly to the 1.5 mer of component A; in this case, the origin of replication was cloned in the pBlueScript II KS+ vector at *Pst* I and *BamH* I sites, and the complete DNA-B was inserted at the *Pst* I restriction site.

## Recombination analysis

Recombination Detection Program version 4 (RDP4) was used for the recombination analysis; this software applies several recombination detections and analysis methods (*Martin et al., 2015*). To search for potential parent viruses in the *nt* database, we used SWeBLAST (*Fourment, Gibbs & Gibbs, 2008*). Then, we aligned the possible parent viruses using the ClustalW algorithm integrated into MEGA v10.0.4 (*Kumar et al., 2018*). Default parameters were used in all the cases.

## Phylogenetic analysis

Evolutionary relationship reconstruction was based on aligning 48 DNA sequences from genomic component A of selected begomoviruses. Based on the Transition Model 3, the maximum likelihood method was used to infer the evolutionary history and count base frequencies directly from the alignment (TIM3+F). The alignment was performed with the MAFFT v7 algorithm with default parameters (*Katoh, Rozewicki & Yamada, 2019*). Evolutionary analyses were carried out using IQ-TREE (*Trifinopoulos et al., 2016*), substitution model was selected automatically by the algorithm. The bootstrap method (1,000 replicates) was applied to test the phylogeny. Visualization and tree edition were carried out in iTOL v5 (*Letunic & Bork, 2021*).
### Supporting data

Supporting data is available in the NCBI database. The metagenome libraries have been deposited at BioProject: PRJNA545097 and Sequence Read Archive (SRA): GCF_018587405.2.

NCBI Reference genome of the species *Sida chlorotic leaf virus*: ASM1858740v2

GenBank accession number for component A of sida chlorotic leaf virus*:* MN013784.1

GenBank accession number for component B of sida chlorotic leaf virus*:* MN013785.1

The new species *Sida chlorotic leaf virus,* described in this work for the first time, has been approved and ratified by the ICTV as reported in: "ICTV Virus Taxonomy Profile: Geminiviridae (2021)" (*Fiallo-Olivé et al., 2021*).

## RESULTS

### Illumina sequencing and sequence assembly of circomics

To detect geminiviruses in positively diagnosed weeds, we conducted a high-throughput sequencing. Restriction enzyme patterns showed that only six weeds (four and two plants from the *Sida* and *Malvastrum* genera, respectively) were infected with geminivirus (Fig. S1). The six positive samples, previously amplified with phi29 DNA polymerase, were pooled and sequenced on the Illumina platform. After sequencing, we obtained 1,273,974 raw reads, of which 974,840 passed the quality filter with an average Phred read quality of 32.28 and an average length of 131 nts. We used the software SPAdes v3.12.0 to assemble the high-quality reads, and then we re-assembled the resulting contigs with the super assembler CAP3. After assembling, 653 contigs longer than 250 bp were obtained, with an average sequence length of 385.57 bp and an N50 of 332.

### Identification of begomovirus in circomics

To annotate the assembled contigs from weed samples, we searched against the *nt* database using the BLASTn tool. The search revealed that 22 contigs got BLAST hits with viruses that belong to the *Begomovirus* genus (Table S2). After curating the 22 contigs, we identified four bipartite BGVs. Three of them belong to previously isolated BGVs species, *Okra yellow mosaic Mexico virus* (OYMV-[MX-COL-10-Si]), *Sida mosaic Sinaloa Virus* (SiMSiV-[MX-Gua-06]), and *Euphorbia mosaic virus* EuMV-[MX-JAL-05-Pep] (Table 1). Component A of the fourth BGV displayed a hit with CuChLV sharing 94.75% of sequence identity (Table 1), a BGV recently identified in a cucumber field in Colima (*Sanchez-Chavez et al., 2020*). Despite the high identity percentage for the fourth BGV, the BLAST hits showed a low query coverage percentage (75%) (Table 1), so we compared the whole genome using MUSCLE alignment. The pairwise comparison indicated that the complete components A of the fourth bipartite BGV and CuChLV shared 86.2% of identity, suggesting that a new species of BGV is present in weed samples.

We performed a comparative genomic analysis to corroborate if component A of the possible new virus is related to CuChLV (BGV that displayed the highest pairwise identity) (Table 1). Comparison among ORFs indicates that CP, REn, and TrAP ORFs shared high pairwise identities (above 93%), while REP and AC4 ORFs and IR shared low pairwise identities (Table S3). Furthermore, a comparison of the entire component A displayed that

**Table 1  Begomoviruses identified in sida and malvastrum plants.**

| BLASTn Hits | Contigs | Component | %Identity | %Coverage | GenBank ID |
|---|---|---|---|---|---|
| CuChLV-[MX-COL-18] | Contig2 | DNA-A | 94.75 | 75 | MN013786.1 |
| | Contig7 | DNA-B | 99.5 | 94 | MN013787.1 |
| EuMV-[MX-JAL-05-Pep] | Contig8 | DNA-A | 98.23 | 100 | DQ520942.1 |
| | Contig6 | DNA-B | 95.7 | 100 | HQ185235.1 |
| OYMV-[MX-COL-10-Si] | Contig10 | DNA-A* | 93.93 | 94 | GU990613.1 |
| OYMV-[MX-COL-10-Her] | Contig4 | DNA-B | 88.6 | 100 | JX219473.1 |
| SiMSiV-[MX-GUA-06] | Contig3 | DNA-A | 93.73 | 99 | DQ520944.1 |
| SiMSiV-[MX-COL-15] | Contig9 | DNA-B* | 94.76 | 95 | MK643154.1 |

**Notes.**

*partial sequences.

the region between nucleotides 61 to 1981 shared 95.8% of pairwise identity, while the part between 1982 to 60 only shared 58.5%. These results suggest that both viruses are related, probably by a recombination event.

Finally, we found a contig (Contig7) that shares 99.5% pairwise identity with component B of CuChLV (MN013787.1). The remaining hits were distributed among plant, insect, and bacterial sequences (Table S2). We confirmed the presence of BGVs components by PCR (Fig. S2).

## Identification of new virus from weed samples

Since component A from the probable new species showed a nucleotide identity below 91%, it meets the criteria to be considered as a new BGV species, so we decided to sequence its genome by Sanger sequencing. First, we identified the weed plants infected by the possible new virus through PCR analysis (F-Rep_Si/R-CP_Si; 1,431 bp, Table S1). The results revealed that all the tested plants (four *Sida* sp. and two *Malvastrum* sp. plants) were infected with the possible new virus (Fig. S2A). To sequence component A of the possible new virus, we extracted the DNA from positively tested plants and amplified by PCR two genomic regions: one of them denominated upper (F-Rep_Si/R-CP_Si; 1,431 bp) and the other one denominated lower region (R-Rep_Si/F-CP_Si; 1,325 bp) (Fig. S3). Amplicons were cloned into pGEM-T easy for later sequencing. Finally, we manually assembled the sequences obtained by Sanger sequencing to get the complete DNA-A sequence of the presumably new virus. The entire DNA-A sequence is 2,635 in length and includes all the typical genomic elements of New World BGVs (Fig. 2). A comparison between DNA-A sequences from Illumina and Sanger sequencing showed that both sequences shared 99% identity, thus corroborating that the sequences obtained by Illumina sequencing are correctly assembled and confirm the possible new virus in weeds. We used the DNA-A sequence obtained by Sanger sequencing for subsequent analyses.

To establish if component A belongs to a new virus, we followed the steps proposed by *Brown et al. (2015)*. First, we selected the first 250 DNA-A sequences resulting from a BLASTn search and aligned them with the MUSCLE algorithm, included in SDT v1.2 software. The demarcation process displayed that the highest pairwise identity reached was 86.4% with CuChLV (Table S4); therefore, this component A belongs to a new BGV

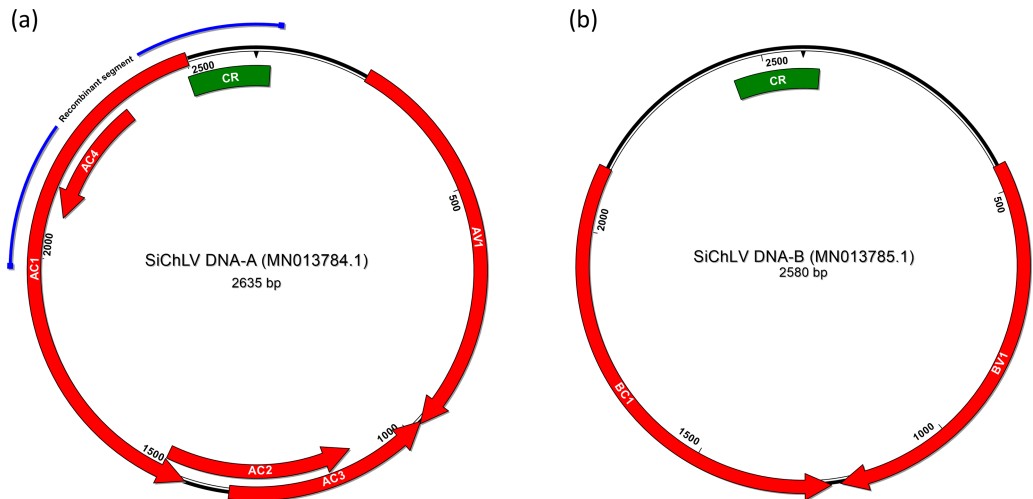

**Figure 2** **Bipartite genome of *Sida chlorotic leaf virus*.** Arrows represent open reading frames of begomoviral genes; the green box signals the common region that harbors the replication origin within components A and B. In DNA-A (A), the blue line indicates the segment that displayed low pairwise identity (58.5%) with CuChLV; a part that probably was acquired from a member of AbMV clade. In the case of component B (B), no signals of a recombination event were detected.

species. Next, we proposed the species name *Sida chlorotic leaf virus* (SiChLV), according to the symptoms observed in sida plants infected by this virus (Fig. 1). We inoculated the new BGV by agroinfiltration in *N. benthamina* to test the infectivity of SiChLV. Plants showed dwarfism and wrinkled leaves with yellow mottling (Fig. S4).

## Identification of CuChLV from weed samples

Since one contig showed high coverage and identity with component B of CuChLV, we decided to search if CuChLV was present in weed samples by PCR amplification using primers designed for component A (F-CuChLV/R-CuChLV; Table S1). PCR amplification showed that one sida plant was infected with CuChLV (Fig. S2B). Interestingly, the plant infected with CuChLV was also infected with SiChLV (Fig. S2A).

## Identification of SiChLV from cucumber samples

Previously, our research group identified CuChLV infecting a cucumber field in Colima, México (*Sanchez-Chavez et al., 2020*); since some sampling sites are close to this cucumber field, we decided to look for SiChLV in the six frozen cucumber samples used in the experiments performed by *Sanchez-Chavez et al. (2020)* by PCR. For the PCR amplification of SiChLV, we used a primer combination for SiChLV that excludes CuChLV (F-Rep_Si/R-CP_Si; 1,431 bp). As a result, we identified three cucumber samples (samples 2, 3 and 4) infected with SiChLV (Fig. S5A). Interestingly, these three cucumber samples were co-infected by CuChLV and SiChLV (Figs. S5A and S5B). Finding SiChLV and CuChLV co-infecting sida and cucumber plants enhance the probability that a recombination event relates to these two viruses.

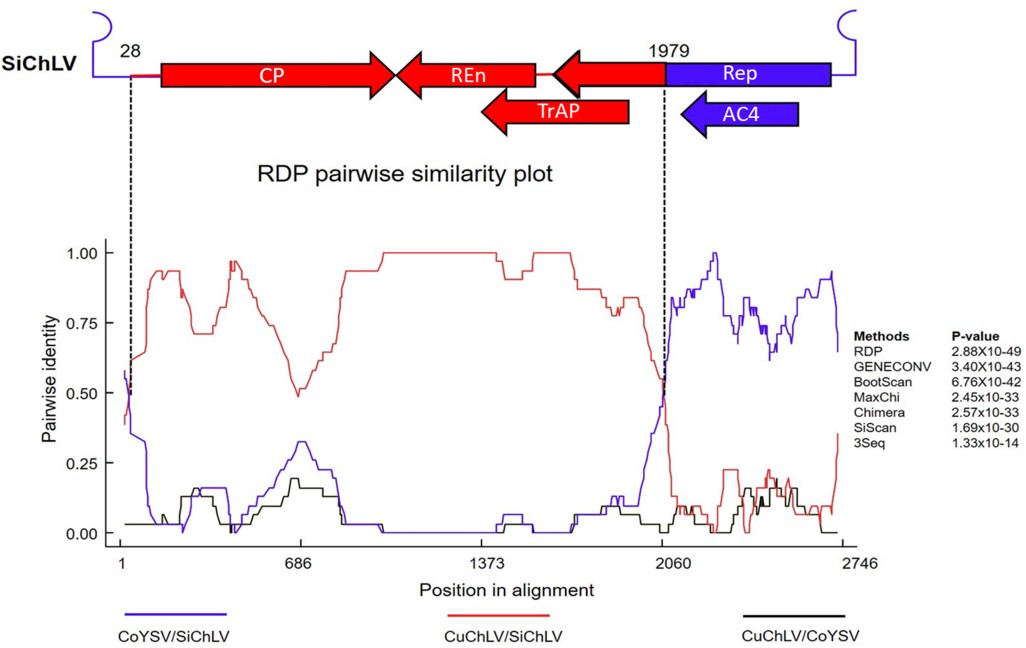

**Figure 3 Recombination analysis of *Sida chlorotic leaf virus*.** Recombination analysis was performed using RDP4. The linear genome map of SiChLV is shown above the RDP pairwise similarity plot. The red region indicates the major parent section inherited from CuChLV. The blue region means the minor parent, donated from a BGV related to the corchorus yellow spot virus (CoYSV). The colored lines in the similarity plot indicate pairwise comparisons between CoYSV and SiChLV (blue), CuChLV and SiChLV (red), and CuChLV and CoYSV (black). Vertical lines indicate the beginning (1979 nt) and ending (28 nt) breakpoints. Filled arrows represent open reading frames (ORFs), and lines symbolize the intergenic region (IR).

## Recombination analysis between CuChLV and SiChLV

Previous observations suggest that a recombination event links CuChLV and SiChLV. To identify possible recombination events between components A of CuChLV and SiChLV, we performed a recombination analysis. First, we searched for potential parent viruses in the GenBank database using SWeBLAST. The search found eight BGVs as the most plausible parents (Table S5). Using this information, we aligned the candidate parents utilizing the MUSCLE algorithm and employed the alignment to feed the RDP4 software. Recombination analysis revealed that SiChLV was originated by recombination; this putative event was supported by seven recombination detection methods, showing *p*-values lower than $1.33 \times 10^{-14}$ (Fig. 3). The major parent for SiChLV was the region between 29-1978 of CuChLV, whereas a 685 nt in size fragment from a BGV belonging to AbMV clade (probably related to corchorus yellow spot virus) serves as the minor parent (Fig. 3). This minor parent fragment includes the entire replication origin (iterons and the hairpin structure with the conserved nona-nucleotide), the entire AC4 ORF, and a 528 nt segment from the REP ORF (Figs. 3 and 2A).

### Phylogenetic analysis of SiChLV

To reconstruct the evolutionary relationship of SiChLV, we performed a phylogenetic analysis using the maximum-likelihood algorithm. The analysis disclosed that SiChLV belongs to the AbMV clade (Fig. 4). Notably, SiChLV is closely related to the cabbage leaf curl Jamaica virus (Fig. 4), a BGV resulting from a probable recombination event between the cabbage leaf curl virus (CabLCV), an SLCV clade member, and a probable member of the AbMV clade. On the other hand, CuChLV, the major parent of SiChLV, was grouped in SLCV clade, as previously reported (*Sanchez-Chavez et al., 2020*).

## DISCUSSION

Evidence provided here supports the role of the *Sida* and *Malvastrum* genera as incubators of new species and as reservoirs of begomoviruses. Scientific reports advised on the potential of weeds as a reservoir or alternative host for BGVs, facilitating their persistence, dissemination to crops of commercial interest, and the emergence of new species (*Jovel et al., 2004*). Weeds from the *Sida* and *Malvastrum* genera are of particular interest because they are reservoirs of several BGVs and natural host of 34 and 10 species, respectively; no other weed genera outnumber these (*Graham, Martin & Roye, 2009*; *Wyant et al., 2011*; *Fiallo-Olivé et al., 2012*; *Tavares et al., 2012*; *Ferro et al., 2017*; *Leke et al., 2020*). The suitability for BGVs infection shown by *Sida* and *Malvastrum,* as well as the ability to harbor BGVs from different species for long periods, could facilitate the co-infection by BGVs, the emergence of new species of begomoviruses arising from recombination, and the spread of BGVs to commercial crops (*Jovel et al., 2004*; *Graham, Martin & Roye, 2009*; *Tavares et al., 2012*; *Ferro et al., 2017*). The results obtained from this work support this premise with the following findings: (1) All sida and malvastrum plant analyses here showed co-infection with two or even three BGVs (Fig. S2). (2) A new virus originated by recombination was identified in sida plants (SiChLV), where CuChLV served as the major parental and one virus from the AbMV clade, probably CoYSV, served as the minor parental (Fig. 3). The recombination event between SiChLV and CuChLV is feasible since co-infections with these viruses were observed (Figs. S2 and S5). Furthermore, a probable similar recombination event between SLCV and AbMV clade members gave rise to CabLCJV, which is closely related to SiChLV (Fig. 4). (3) SiChLV displays the ability to infect non-cultivated plants (sida and malvastrum) and cultivated plants (cucumber) (Fig. S5). This ability is common in BGVs identified in *Sida* genus plants; at least nine species of BGVs first discovered in sida plants could jump to commercial crops, such as soybean (*Fernandes et al., 2009*), bean (*Durham et al., 2010*), tomato (*Rocha et al., 2013*), among others (Table S6). Here we documented that a BGV identified in the S*ida* genus spread and infected a cucumber crop.

Another interesting aspect of the *Sida* and *Malvastrum* genera is that they act as virus reservoirs whose original hosts are plants from commercial crops (Tables S6 and S7) (*Idris & Brown, 2002*; *Chowda Reddy et al., 2005*). Here we found sida and malvastrum plants harboring viruses that infect okra (OYMMV) and cucumber (CuChLV) (Figs. S2B and S2D); this supports previous reports of sida and malvastrum plants serving as a reservoir of BGVs that affect commercial crops. The findings published here allow us to recommend

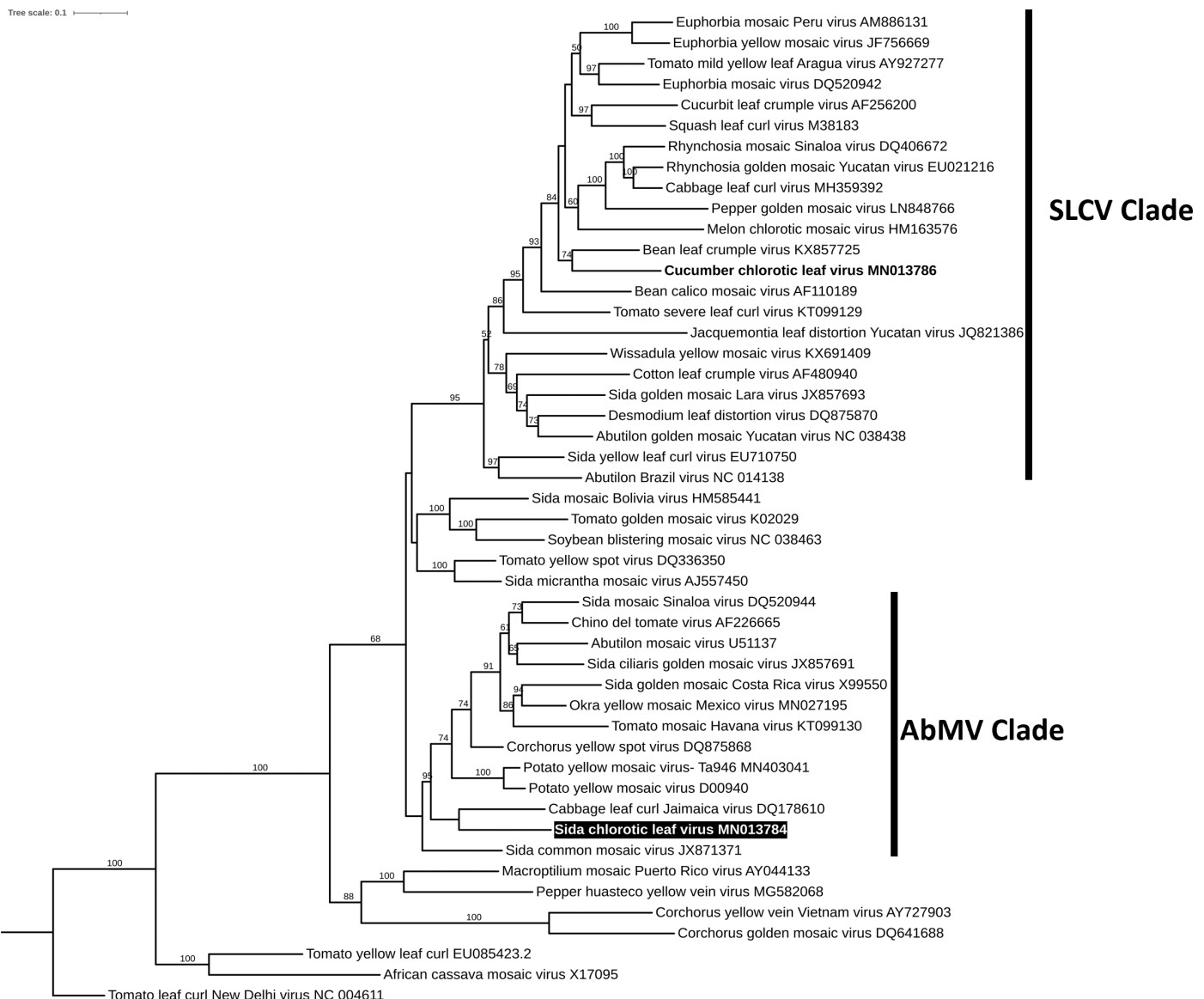

**Figure 4   Phylogenetic analysis of Sida chlorotic leaf virus.** Phylogenetic relationships of the novel virus (highlighted) were based on the alignment of the full DNA-A sequences of 48 selected begomoviruses. Phylogenetic trees were inferred using the maximum likelihood method based on the Transition model 3 and count base frequencies directly from the alignment (TIM3+F). Bootstrap values (1,000 iterations) are indicated for each node. GenBank accession numbers are located after the species name.

that weeds surrounding field crops, especially those from genera *Sida* and *Malvastrum*, should be removed to eliminate or reduce sources of BGVs, which directly affect the amount and quality of crops and bring about critical economic losses. Additionally, our work demonstrated that Colima is rich in begomovirus species. This Mexican state is an important producer (one-third of its territory is intended for agriculture *Agricultural and Fisheries Information Service, 2018*). Exporter of cherry tomato, coconut, papaya, banana, cucurbits, and lemon (*Agricultural and Fisheries Information Service, 2018*). 11

BGVs species are found in Colima, representing about 30% of all the species isolated in México. The number of BGVs found in this small Mexican state granted the second place of BGV richness in México, just behind Yucatán (18 species). The begomovirus richness observed in Colima may be due to its climate. Regions with a dry season with four months of less than 80 mm rainfall, and a mean monthly temperature in the hottest month not below 21 °C facilitate *B. tabaci* distribution (*Seal, VandenBosch & Jeger, 2006*). The subhumid climate of Colima facilitates the distribution of *B. tabaci* (*Instituto Nacional de Estadística y Geografía , 2016*). The richest Mexican states (Yucatán, Colima, and Sinaloa) in BGVs species share very similar climate patterns (*Instituto Nacional de Estadística y Geografía , 2017a*; *Instituto Nacional de Estadística y Geografía , 2017b*).

## CONCLUSIONS

Our work strengthens the fact that plants from the *Sida* and *Malvastrum* genera play an important role in the safety of commercial crops as they can host multiple BGVs species and supports the role of weeds as a critical biotic factor affecting the production of crops. Furthermore, the suitability displayed by these genera favors events, such as recombination, that could lead to the emergence of new species or varieties of BGVs with the ability to affect plants that they could not before. For example, as seen in this study, a new begomovirus (SiChLV) emerged by recombination and was identified in non-cultivated plants, gaining the aptness to spill over to cucumber plants. Finally, our conclusions can promote awareness among agricultural producers about the importance of controlling these weeds, not only to avoid their harmful direct effects or to control pests but also as a preventive measure to reduce the risk of facing negative consequences associated with BGV infections that threaten agricultural efficiency and nutritional food security.

## ACKNOWLEDGEMENTS

We would like to thank the National Laboratory LANBAMA and LANGEBIO for the technical assistance provided.

### Funding

This research was funded by the Consejo Nacional de Ciencia y Tecnología (CONACYT) of México. Grant number APN-2015-01-741 to Yair Cárdenas-Conejo. Daniel Alejandro García-Rodríguez and Brenda L. Palacios-Partida were supported by a fellowship from the Consejo Nacional de Ciencia y Tecnología (CONACYT, México). Carlos Fernando Regla-Márquez was supported by a fellowship from the Programa de Mejoramiento del Profesorado (PROMEP). The funders had no role in study design, data collection and analysis, decision to publish, or preparation of the manuscript.

### Grant Disclosures

The following grant information was disclosed by the authors:

Consejo Nacional de Ciencia y Tecnología (CONACYT) of México: APN-2015-01-741. Programa de Mejoramiento del Profesorado (PROMEP).

## Competing Interests

The authors declare there are no competing interests.

## Author Contributions

- Daniel Alejandro García-Rodríguez conceived and designed the experiments, performed the experiments, prepared figures and/or tables, and approved the final draft.
- Brenda Lizet Partida-Palacios performed the experiments, prepared figures and/or tables, and approved the final draft.
- Carlos Fernando Regla-Márquez performed the experiments, prepared figures and/or tables, and approved the final draft.
- Sara Centeno-Leija analyzed the data, authored or reviewed drafts of the article, and approved the final draft.
- Hugo Serrano-Posada analyzed the data, authored or reviewed drafts of the article, and approved the final draft.
- Bernardo Bañuelos-Hernández analyzed the data, authored or reviewed drafts of the article, and approved the final draft.
- Yair Cárdenas-Conejo conceived and designed the experiments, analyzed the data, authored or reviewed drafts of the article, and approved the final draft.

## Field Study Permissions

The following information was supplied relating to field study approvals (i.e., approving body and any reference numbers):

The samples were collected in public areas such as roadsides and uncultivated lands. No permission to access those places was needed.

## DNA Deposition

The following information was supplied regarding the deposition of DNA sequences:

The genome of Sida chlorotic leaf virus are available in NCBI: DNA-A: MN013784.1, DNA-B: MN013785.1.

## Data Availability

The raw data, the unedited figures related to gel electrophoresis from PCR assays and sequencing data, is available in the Supplemental Files.

## New Species Registration

The following information was supplied regarding the registration of a newly described species:

This new species *Sida chlorotic leaf virus* has been approved and ratified by the ICTV as reported in: "ICTV Virus Taxonomy Profile: Geminiviridae (2021)". The Journal of General Virology 102:1696. DOI: 10.1099/JGV.0.001696.

## Supplemental Information

Supplemental information for this article can be found online at http://dx.doi.org/10.7717/peerj.15047#supplemental-information.

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
