# Peer review of "Sida chlorotic leaf virus: a new recombinant begomovirus found in non-cultivated plants and Cucumis sativus L"

_PeerJ, doi:10.7717/peerj.15047_

## Round 0.1 · original submission · Major Revisions

Please provide a point by point response to reviewers and resubmit the updated version of the manuscript.

Reviewer 1 ·

Basic reporting

This manuscript shows that the new begomovirus (recombinant) were identified in both non-cultivated crops and Cucumis sativus L. It contains new content in that a novel mutant called Sida chlorotic leaf virus has been identified.

Experimental design

Even though authors mentioned "interspecies transmission" expressed in the title, the part about how this virus was actually transmitted was omitted. For this reason, it is difficult to find the information expected through the title in the manuscript. The overall design of the experiment seems to be well done, but I suggest modifying the title.

Validity of the findings

Although information on recombinant viruses was systematically presented using various bioinformatics techniques, it was mainly found in cases with multiple infections, so it needs to check what virus(es) makes symptoms authors observed. I thought that this manuscript should be evaluated again after a few corrections including the followings.

Additional comments

Line 72, Bemisia Tabaci --> Bemisia tabaci

Lines 81-82, it is hard to understand "in the cultivo cultivate"

Line 90, "native to Africa, India, Asia, and Japan" --> Continents and countries are mixed. Please change it.

Line 121, "where the Tomato yellow leaf curl virus cause the loss of crops" --> tomato yellow leaf curl virus (not Italic)

Lines 151-152, names of restriction enzyme --> Italic

Line 153, 1000 to 3500 --> 1,000 to 3,500

Line 184, 95 °C --> 95°C, 56 °C --> 56°C

Line 194, taatatt/a1c --> TAATATT/AC what is "1" between "a" and "c"?

Lines 250, 282, 311, Muscle alignment --> MUSCLE alignment

Line 300, 1431 bp --> 1,431 bp

Line 348, "An interspecies transmission of SiChLV between non-cultivable plants (sida and malvastrum) and cultivable plants (cucumber) was identified (Figure S4)."
--> Although it has been identified in individual plants, it has not been confirmed that it is actually transmitted between plants, so be careful in the choice of words.

Line 391, The following are --> The followings are or The following is

Fig. 5 specie --> species

Reviewer 2 ·

Basic reporting

The manuscript by Garcia-Rodriguez et al. describes the detection and characterization of a new recombinant begomovirus in Sida and Malvastrum sp. in Mexico, which they name Sida chlorotic leaf virus (SiChLV). The virus was initially detected by HTS and was subsequently cloned and Sanger-sequenced. Interestingly, the plants had mixed infections with other begomoviruses, including one (CuChLV) that seems to be the major parent of the new virus. The authors also showed that some of the previously collected cucumber samples from which CuChLV was isolated were co-infected with SiChLV. The fact that both viruses were found in both Sida and cucumber samples is really interesting, indicating the potential of SiChLV of spilling-over from uncultivated to cultivated plants, which is a prerequisite for the emergence of new pathogens. Thus, the manuscript does contain valuable information which merits publication. However, there are a number of major issues (both general and specific) that must be addressed before it can be accepted. General issues: (i) quality of the language must be improved, (ii) most of the figures (but in particular Figs. 2 and S2) must also be improved, (iii) figure and table legends must be self-explanatory, (iv) many papers are cited inappropriately, with the relevant papers not being cited. Specific comments are provided below.

Specific comments

L143: How were the plants identified ? It is very difficult to identify malvaceous plants even at the genus level by visual observation. They should be identified by DNA barcoding.

L147: Which modifications ?

L149: phi29 is a bacteriophage. What was used was the phi29 DNA polimerase (same in L233). What was the protocol used for RCA ? There are at least three possibilities: (i) the TempliPhi kit, according to the manufacturer's instructions; (ii) Inoue-Nagata et al., J Virol Meth 2004; (iii) Haible et al., J Virol Meth 2006.

L149-154: My interpretation of these sentences is that, prior to HTS, the authors screened the samples by RCA-RFLP to make sure they only sequenced begomovirus-positive samples. For this, RCA products were digested with the restriction enzymes listed and the resulting band pattern was analyzed. Patterns indicative of the presence of a begomovirus were those in which band sizes added up to multiples of 2,600 nt. Assuming that this interpretation is correct, the text needs to be rewritten to clearly indicate this. And, of course, it should also be modified accordingly in case my interpretation is incorrect.

L157: I am surprised that the authors did not barcode their samples to be able to trace the reads back to each individual sample (and I was even more surprised when I read in L231-232 that only six samples were positive for the presence of a begomovirus, which would have made barcoding extremely efficient).

L230-231: What was used for diagnostics was RCA-RFLP, *not* HTS. The restriction patterns should be provided as a supplementary figure.

L238: The text states that 653 contigs were obtained, but Table S2 lists 647 contigs. Which one is correct ?

L246-250: Again, I can't reconcile this with Table S2. The problem is that the sequences of SiChLV have been deposited in GenBank, and Table S2 lists these sequences as hits (but then, how can the authors explain that the contig 2 displays only 96% identity and 83% coverage with SiChLV DNA-A ? It should be 100% identity and 100% coverage. Same for contig 203 and SiChLV DNA-B). Ideally, Table S2 would not list SiChLV as a hit, so that the main text and the Table S2 inform the same results. As for Table 1, first of all it doesn't make sense to have a column listing the samples as "weeds". This column should be deleted, and replaced by as column listing the contig number for each hit. Second, there's almost zero correspondence between identity and coverage values in Tables 1 and S2 (and why no values for SiMSiV in Table S2 ?). Third, the accession number MN013784 belongs to SiChLV DNA-A, *not* CuChLV DNA-A (that would be MN013786). Fourth, Table 1 does not list a CuChLV DNA-B hit, although one is listed in Table S2 (contig 7). In summary, there are multiple major problems with the tables that make it essentially impossible to reconcile the main text, Table 1 and Table S2. The information provided in these three places must match 100%. Three things to fix: (i) Table 1 should be a subset of Table S2, listing the contig number that matched each begomovirus component; (ii) Table S2 should not list SiChLV as a hit, instead informing CuChLV as a hit (so that the information matches that of the main text and of Table 1); (iii) double-check all accession numbers in both tables.

L263: Figure S1c is missing the lane markers. The legend of this figure needs to provide all the information required for the figure to be interpreted (this applies to all supplementary figures and table). For example, which viruses/components are being detected in each gel ? Is it correct that no positive controls were used ? (I'm assuming, since there's no lane marked as "P"; if my assumption is correct, then these tests should be repeated with the inclusion of the appropriate positive controls)

L278: This is actually quite significant: 1% of a 2600 nt genome means that there are approx. 26 nucleotide differences between the Illumina- and Sanger-derived sequences. Considering the higher error rate of Illumina compared to Sanger (plus the fact that Sanger sequences can be checked manually), I'm assuming that the authors used the Sanger-derived sequence in all of their subsequent analyses ? This should be informed in the manuscript.

L286-288: There's nothing in the M&M about infectious clones and agroinoculation. How were the infectious clones obtained ? What was the protocol used for agroinoculation ? This is essential information that must be provided.

L290: I'm curious as to what is the percent identity between SiChLV and CuChLV DNA-Bs.

L330-336: The correct citation is Jovel et al., 2004 (also in L340). Please do *not* cite Prajapat, Marwal & Gaur, 2014. This "review" is so plagued by gross mistakes (such as begomoviruses infecting monocots) that it should have been retracted. At the very least, it should never be cited so that its mistakes are not propagated. Also, the ICTV taxonomy profile of the family Geminiviridae, cited here, is most definitely not the appropriate reference for the previous (essentially correct) sentences. Instead, there's a very large number of papers describing Sida and other malvaceous species as natural reservoirs of begomoviruses. A non-exhaustive list includes: Leke et al., Arch Virol. 165:775-779, 2020; Ferro et al., Ann Appl Biol 170:204-218, 2017; Fiallo-Olivé et al., Arch Virol 160:3161-3164, 2015; Tavares et al., Planta Dan 30:305-315, 2012; Fiallo-Olivé et al., Arch Virol 157:141-146, 2012; Wyant et al., Arch Virol 156:347-352, 2011; Graham et al., Virus Genes 40:256-266, 2010; Guo& Zhou, Virus Genes 33:279-285, 2006; Jovel et al., Arch Virol 149:829-841, 2004; Roye et al., Plant Dis 81:1251-1258, 1997; Hofer et al., Virology 78:1785-1790, 1997; Frischmuth et al., J Gen Virol 78:2675-2682, 1997.

L341: These are indeed relevant findings, but they are hardly "remarkable". Mixed infection of Sida and other malvaceous species by begomoviruses have been reported in the literature for decades. Likewise, the notorious recombinogenic nature of begomoviruses has been recognized since at least 1999 (Padidam et al.).

L363-369: Most likely, this simply reflects that these are places where researchers are actively looking for these viruses.

Experimental design

Please see section 1.

Validity of the findings

Please see section 1.

---

## Round 0.2 · Minor Revisions

Dear Authors,

I have received the second round revision of your manuscript. You have only to revise your paper following very few suggestions this time.

Thank you
Giuseppe Parrella

Reviewer 1 ·

Basic reporting

The manuscript has been revised by faithfully reflecting the points pointed out by reviwers last time. I hope that this paper will be accepted after the following minor corrections are made.

- When writing the name of a restriction enzyme, only the first three letters derived from the scientific name should be written in italics, and the rest should not be written in italics.

- Please enter the company name for "pBlueScript II KS+ vector".

- Check ICTV for how to write scientific names for viruses.

Experimental design

No problem was found in this time.

Validity of the findings

No problem was found in this time.

Additional comments

No problem was found in this time.

Reviewer 2 ·

Basic reporting

The revised version is greatly improved. My comments and suggestions were either followed or, when not followed, very appropriately rebutted (I particularly liked the way the authors explained how they identified Malvastrum and Sida plants). I would use "non-cultivated" in the title. I don't think "non-cultivable" is grammatically correct (but English is not my native language so I'm not 100% sure). I believe the manuscript can now be accepted for publication. Note that there's a small number of spelling errors (for example, Malvastum in the legend of Figure 1) but these can be corrected during the production stage.

Experimental design

Details of the RCA and, most important, of the construction of infectious clones and the agroinoculation procedure, are now provided. One comment regarding the barcoding of samples for NGS. The authors say that they did not use barcoding because it would increase costs. This is really not true. Although adding the barcodes does increase the cost of library construction, the total cost is much lower because barcoded libraries are pooled for sequencing, thus using a fraction of the flow cell(s). The very reason to barcode samples is to reduce sequencing costs.

Validity of the findings

As mentioned in my previous review, the manuscript does contain interesting information which merits publication.

Additional comments

Not really related to the manuscript, but more as a friendly suggestion for the authors' future work. I asked what were the changes in the Dellaporta protocol, and the authors explained that they added a phenol/chloroform step. That's ok as far as this manuscript is concerned, and I certainly agree that adding this step will increase the efficiency of the method. However, it basically defeats the purpose of the Dellaporta procedure - it's biggest advantage is that it does not include a phenol step ! (I think there are very few chemicals that are more dangerous than phenol in a lab environment) Now, in my lab we also had problems with Dellaporta extraction when working on weeds, and especially when working with dried leaf material. In these cases we used the protocol from Doyle and Doyle (A rapid DNA isolation procedure for small amounts of fresh leaf tissue. Phytochemical Bulletin 19:11-15, 1987). This procedures uses non-toxic CTAB instead of phenol, and in our hands yields high amounts of high-quality DNA, even when working with weeds and/or dried samples.

---

## Round 0.3 · accepted · Accept

Thank you for submitting your recent version of the manuscript and congratulation on its acceptance.